# Cesium-mediated electron redistribution and electron-electron interaction in high-pressure metallic CsPbI$_3$

Feng Ke [1,2], Jiejuan Yan[2], Shanyuan Niu[1,2,7], Jiajia Wen [1], Ketao Yin [3] ✉, Hong Yang [2], Nathan R. Wolf [4], Yan-Kai Tzeng[5], Hemamala I. Karunadasa [1,4], Young S. Lee [1,6], Wendy L. Mao [1,2] & Yu Lin [1] ✉

Electron-phonon coupling was believed to govern the carrier transport in halide perovskites and related phases. Here we demonstrate that electron-electron interaction enhanced by Cs-involved electron redistribution plays a direct and prominent role in the low-temperature electrical transport of compressed CsPbI$_3$ and renders Fermi liquid (FL)-like behavior. By compressing δ-CsPbI$_3$ to 80 GPa, an insulator-semimetal-metal transition occurs, concomitant with the completion of a slow structural transition from the one-dimensional *Pnma* (δ) phase to a three-dimensional *Pmn*2$_1$ (ε) phase. Deviation from FL behavior is observed upon CsPbI$_3$ entering the metallic ε phase, which progressively evolves into a FL-like state at 186 GPa. First-principles density functional theory calculations reveal that the enhanced electron-electron coupling results from the sudden increase of the 5d state occupation in Cs and I atoms. Our study presents a promising strategy of cationic manipulation for tuning the electronic structure and carrier scattering of halide perovskites at high pressure.

Three-dimensional (3D) halide perovskites of the form ABX$_3$ (A = organic or inorganic cation, B = metal cation, and X = halide anion) display remarkable optical and electronic properties that find applications for a wide range of technologies such as photovoltaics[1,2] and light-emitting diodes[3]. In these materials, BX$_6$ octahedra corner-share to form the 3D perovskite structure. Beyond the perovskite phase, BX$_6$ octahedra can also connect through face- and edge-sharing and extend into 1D and 2D structures. At ambient conditions, the electronic states of the metal-halide sublattice are typically responsible for the valence band maximum (VBM) and conduction band minimum (CBM) of these ABX$_3$ phases, and govern their optical absorption[4–6]. It is believed that the A-site cation in ABX$_3$ phases only makes a minimal

contribution to the band edge states and plays an indirect role in the electronic structures of these materials[7,8].

Modulating the electronic structures and searching for intriguing electronic states in ABX$_3$ phases have attracted considerable experimental and theoretical attention, partly motivated by their structural similarity to the perovskite oxides where exciting electronic states such as ferroelectricity[9], unconventional superconductivity[10,11], and topological insulating behavior[12] have been discovered. Recent experimental studies observed ferroelectric-like behavior in (MA)PbI$_3$ (MA = CH$_3$NH$_3$$^+$) at ambient pressure, although this topic is still under debate[13–15]. Compositional tuning or designing lattice architectures are effective methods for modifying their electronic structures and carrier

[1]Stanford Institute for Materials and Energy Sciences, SLAC National Accelerator Laboratory, Menlo Park, CA 94025, USA. [2]Department of Geological Sciences, Stanford University, Stanford, CA 94305, USA. [3]School of Physics and Electronic Engineering, Linyi University, Linyi, Shandong 276005, China. [4]Department of Chemistry, Stanford University, Stanford, CA 94305, USA. [5]Department of Physics, Stanford University, Stanford, CA 94305, USA. [6]Department of Applied Physics, Stanford University, Stanford, CA 94305, USA. [7]Present address: College of Engineering and Applied Sciences, Nanjing University, Nanjing, Jiangsu 210093, China. ✉e-mail: yinketao@lyu.edu.cn; lyforest@stanford.edu

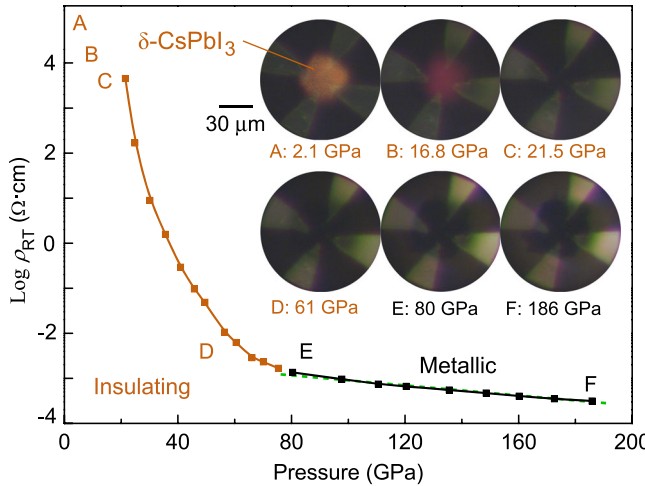

**Fig. 1 | Room-temperature resistivity (Log $\rho_{RT}$) of CsPbI$_3$ as a function of pressure.** The orange and black squares represent the data from the insulating and metallic phases, respectively. The green dashed line is a linear fit of the log $\rho_{RT}$–$P$ curve above 80 GPa. A slope of $-(5.3 \pm 0.4) \times 10^{-3}$ Ω·cm/GPa is obtained. The photomicrographs show the hand-wired electrode configuration for the high-pressure resistivity measurements and the color change of the sample from yellow to red and further to opaque black under compression.

transport properties[16]. Lattice compression has also been found to be a useful means for accessing different electronic states[17–25]. Metallic-like behavior was experimentally achieved in compressed (MA)PbI$_3$, (FA) PbI$_3$ (FA = CH(NH$_2$)$_2$$^+$), and δ-CsPbI$_3$[26–28]. Bandgap closure and reopening were observed in the double perovskite Cs$_2$Au$_2$I$_6$ at high pressure[29]. However, almost all these high-pressure experimental efforts were conducted near room temperature. Theoretically, it is predicted that under compression halide perovskites could be topological insulators—e.g., cubic CsPbI$_3$[30–32], or superconductors—e.g., Cs$_2$Tl$_2$X$_6$[33].

At low temperatures, correlated materials can display fascinating phenomena such as various types of metal-insulator and superconducting transitions due to strong electron–electron (e-e), electron–phonon (e-p), and electron–magnon interaction. The carrier transport and scattering mechanisms have well-known temperature dependences. For instance, in simple metals the e-p interaction plays an important role in the carrier transport, leading to a $T^5$ dependence of resistivity at low temperatures with a linear temperature dependence above the Debye temperature[34]. For interacting fermion systems, the carrier scattering has been well described by the Fermi liquid (FL) model[35], in which carriers are considered as quasiparticles and the e-e interaction dominates the low-temperature electrical transport, causing a $T^n$ ($n = 2$) dependence of resistivity for 3D metals. In certain correlated materials, FL behavior has been observed to break down, resulting in non-Fermi liquid (NFL) behavior with $n < 2$, e.g., in the normal state of high-temperature iron- and cuprate-based superconductors[36–39]. ABX$_3$ phases were reported to have a Debye temperature of 100–250 K[40,41]. Hence carrier transport behavior at low temperature will be different from that at above the Debye temperature. At ambient pressure, e-p coupling was believed to be mainly responsible for the scattering of the free charge carriers in ABX$_3$-based materials[42–44], although a recent study indicated that e-e interaction had contributions to the low-energy excitons in semiconducting (FA) PbI$_3$ at ambient pressure[45]. Studying the electronic structures and carrier scattering mechanisms in these materials at low temperature combined with high-pressure conditions represents unexplored territory.

In this work, we study the electronic states in CsPbI$_3$ over a vast pressure-temperature space of 0.1–186 GPa and 2–300 K. With

application of pressure to 80 GPa, insulating δ-CsPbI$_3$ transforms to a semimetallic and then to a metallic state, accompanied by the completion of a sluggish $Pnma$-to-$Pmn2_1$ (δ-to-ε) phase transition from a 1D chain structure to a 3D structure. Deviation from a FL-like state is observed as CsPbI$_3$ enters the metallic ε phase. With further compression to 186 GPa, the deviation gradually disappears, and the metallic phase behaves as a Fermi liquid. Band structure and orbital occupation calculations reveal sudden increase of the Cs- and I-5d state occupation in the ε phase that strengthen the e-e interaction and induce the FL-like behavior. In contrast to ambient conditions where the A-site cation only has an indirect effect on the electronic properties of the ABX$_3$ phases, the Cs cation plays a direct and prominent role in the electronic structure of CsPbI$_3$ at high pressure.

## Results

We measured the resistivity of CsPbI$_3$ over a temperature range of 2–300 K and pressures up to 186 GPa (Figs. 1 and 2). Sample characterization and pressure calibration details can be found in the supporting information (Supplementary Figs. 1–4). At ambient conditions, the room-temperature resistivity ($\rho_{RT}$) is beyond the measurable range of the instrument since δ-CsPbI$_3$ is a large bandgap (2.5–2.8 eV) insulator[46,47]. With increasing pressure, the value of $\rho_{RT}$ falls into the measurable range and is $4.5 \times 10^3$ Ω·cm at 21.5 GPa, accompanied by the color of the sample changing from yellow to red and then to black. With further compression to 80 GPa, $\rho_{RT}$ drops dramatically by more than six orders of magnitude, beyond which a linear reduction of log $\rho_{RT}$ at a rate of $(5.3 \pm 0.4) \times 10^{-3}$ Ω·cm/GPa continues up to 186 GPa, suggesting an electronic transition at 80 GPa. During decompression, $\rho_{RT}$ at ~20 GPa is comparable with the values during the compression process (Supplementary Fig. 5). Upon further decompression, $\rho_{RT}$ is beyond the measurable range again, followed by the color of CsPbI$_3$ changing from black to red and then to yellow. During recompression, $\rho_{RT}$ decreases with a similar trend to the first compression cycle.

The resistivity–temperature ($\rho$–$T$) curves of CsPbI$_3$ at representative pressures are shown in Fig. 2. At pressures below 50 GPa, the temperature dependence of resistivity (d$\rho$/d$T$) was negative throughout the temperature range measured (2–300 K), indicating an insulating character. By fitting the linear range of the ln $\rho$–1/$T$ curves at high temperature to the Arrhenius equation, an activation energy of $134.8 \pm 5.3$, $59.1 \pm 2.1$, and $29.5 \pm 0.7$ meV is obtained at 30, 41, and 50 GPa, respectively. Upon further compression to 61 GPa, the resistivity starts to decrease with cooling (d$\rho$/d$T$ > 0, metallic behavior) near room temperature but reverses its temperature dependence back to d$\rho$/d$T$ < 0 below ~255 K, consistent with a semimetallic state. The transition temperature shifts to ~135 K at 70 GPa. A linear extrapolation suggests that at a critical pressure of 80.5 GPa, the sign reversal of d$\rho$/d$T$ at low temperature could be fully suppressed. The experimental result at 80 GPa indeed shows a positive d$\rho$/d$T$ throughout the temperature range of 2–300 K, indicating the electronic transition to a metallic state in compressed CsPbI$_3$. A previous study observed metallic-like behavior in CsPbI$_3$ near room temperature at ~40 GPa[28]. However, the limited temperature range, missing experimental details, and their inconsistent structural data of the metallic-like phase make direct comparison difficult.

Detailed analysis of the $\rho$–$T$ curves reveals remarkable behavior in the metallic phase. For $T > 110$ K, a linear temperature dependence is obtained in all the $\rho$–$T$ curves above 80 GPa (Fig. 2a), which is consistent with the well-known simple metal model where the e-p interaction dominates the carrier transport. The $\rho$–$T$ relationship obeys the following equation,

$$\rho(T) = \rho_0 + BT^5 + CT \tag{1}$$

where $\rho_0$ is the residual resistivity, and B and C are the coefficients of $T^5$ and $T$-terms, respectively. Surprisingly, the $\rho$–$T$ curves below 110 K

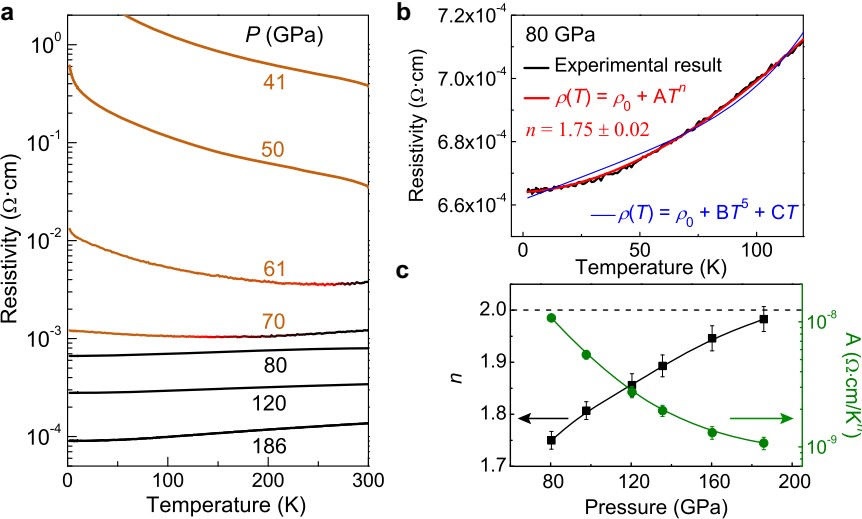

**Fig. 2 | Low-temperature electrical transport properties of CsPbI₃ at high pressure. a** Resistivity–temperature ($\rho$–$T$) curves at representative pressures ($P$) showing the insulating-semimetallic-metallic transition in compressed CsPbI₃. The orange and black lines represent the insulating and metallic phases, respectively. The red lines in 61 and 70 GPa mark the transition temperatures where $d\rho/dT$ changes from negative to positive values. **b** Fitting of the $\rho$–$T$ curve at 80 GPa to the models of $\rho(T) = \rho_0 + AT^n$ (red line) and $\rho(T) = \rho_0 + BT^5 + CT$ (blue line). **c** The exponent $n$ (black squares) and coefficient $A$ (green circles) obtained using the $\rho(T) = \rho_0 + AT^n$ model as a function of pressure. Error bars are from the fitting of the $\rho$–$T$ curves.

significantly deviate from this $T$-dependence (Fig. 2b, and Supplementary Fig 6). Instead, $\rho(T)$ is found to be proportional to $T^n$,

$$\rho(T) = \rho_0 + AT^n \qquad (2)$$

where the fitted exponent $n$ is $1.75 \pm 0.02$ at 80 GPa and increases gradually with pressure to $1.98 \pm 0.03$ at 186 GPa (Fig. 2c). The observation of $\rho(T) \sim T^n$ where $n \le 2$ in compressed CsPbI₃ at low temperature is anomalous. Such behavior has typically been observed in materials such as high-temperature superconductors and heavy fermion compounds[36–39], in which the d or f electron shell is partially filled and e-e interaction is important. However, at ambient conditions δ-CsPbI₃ does not have any d-electrons that can readily serve as charge carriers.

To understand the abnormal low-temperature behavior of the high-pressure metallic phase, we studied the structural evolution of δ-CsPbI₃ as a function of pressure using powder X-ray diffraction (XRD) measurements (Fig. 3a and Supplementary Figs. 7–14). Below 5.4 GPa, δ-CsPbI₃ crystallizes in a 1D double chain structure with a *Pnma* symmetry where PbI₆ octahedra edge-share along the b-axis (Supplementary Fig. 10), consistent with the reported structure at ambient conditions[28,48]. Upon compressing to 6.7 GPa, new diffraction peaks (marked as red arrows in Fig. 3a and Supplementary Fig. 7) start to appear and more peaks develop with further compression, indicating the emergence of a high-pressure phase (ε-CsPbI₃). This δ-to-ε structural transition is found to be very sluggish where the high-pressure phase grows at the expense of the low-pressure phase. For instance, the peak that appears at ~9.8° (red triangle in Fig. 3a) at 6.7 GPa rapidly increases its intensity and becomes the strongest peak above 20.3 GPa, accompanied by a dramatic intensity decrease of the (211) and (212) diffraction peaks of the low-pressure phase. The (211) and (212) reflections are the most intense peaks of the starting structure and eventually disappear by 82 GPa, indicating the completion of the phase transition that spans a pressure window of ~75 GPa. With further compression up to 184 GPa (Supplementary Fig. 9), the XRD patterns are similar except for peak shifts to higher diffraction angles and subtle changes in the peak widths and intensities. Under decompression, CsPbI₃ goes back to the initial δ phase after releasing pressure. Between 6.7 and 82 GPa, new reflections that can be indexed into the

high-pressure ε phase gradually appear upon compression, and the relative peak intensities of the δ phase change with pressure. These observations indicate that the structures of the δ and ε phases evolve within this pressure range. Rietveld refinement indicates that an orthorhombic structure with a *Pmn*2₁ space group fits the diffraction patterns above 82 GPa well (Supplementary Fig. 11). In contrast to the initial 1D double chain structure, the high-pressure ε phase is a 3D structure with adjacent double chains bonding together where the Pb atoms are eight- and nine-fold coordinated by the I atoms (Fig. 3c and Supplementary Fig. 10).

Comparing the electrical transport and XRD results, we find that the insulator-semimetal-metal transition and the *Pnma*-to-*Pmn*2₁ phase transition are closely related. The pressure at which the structural transition completes coincides with the end pressure of the metallic transition and with that of the change in the $\rho_{RT}$ – $P$ curve, i.e., ~80 GPa. Further analysis of the lattice parameters of the *Pmn*2₁ phase show pronounced deviations between 62 and 82 GPa (Supplementary Fig. 12), especially for the $a$ lattice constant. This pressure window is consistent with that of the metallic transition which starts at ~61 GPa where CsPbI₃ shows a metallic character above ~255 K and completes at ~80 GPa where a metallic state is present throughout 2–300 K temperature range. Above 85 GPa, the lattice parameters (Supplementary Fig. 12) and the unit cell volume (Supplementary Fig. 13) of the *Pmn*2₁ phase change smoothly as a function of pressure and no obvious anomaly is observed.

First-principles density functional theory (DFT) calculations corroborate the experimental results (Fig. 3b). We performed extensive structural searches on CsPbI₃ using the CALYPSO software package, which can predict energetically stable structures of materials at extreme conditions, e.g., high pressure[49]. At 90 GPa, the structural searches find that an orthorhombic *Pmn*2₁ phase and two monoclinic structures with *Cm* and *C*2/*m* symmetries are energetically stable and have lower enthalpies than the starting *Pnma* δ-CsPbI₃. Among these structures, only the *Pmn*2₁ structure produces a simulated XRD pattern that is consistent with the experimental pattern (Supplementary Fig. 11). After structure optimization, the detailed structural parameters of the *Pmn*2₁ phase agree well with those obtained from XRD refinement (Supplementary Table 1). The difference in the transition pressure between experiments (6.7 GPa, Fig. 3a and Supplementary

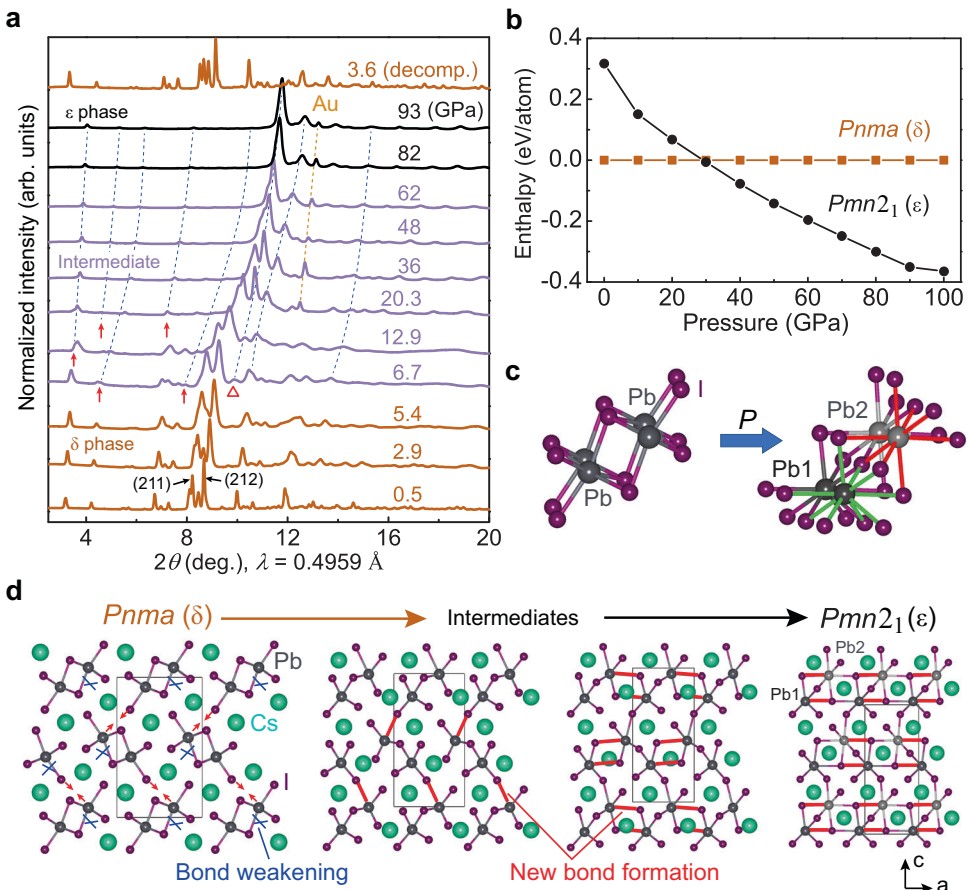

**Fig. 3 | Experimental and theoretical evidence for the pressure-induced δ-to-ε structural transition in CsPbI₃. a** Selected XRD patterns of compressed CsPbI₃ showing the gradual δ-to-ε phase transition. See Supplementary Fig. 7 for the full set of XRD patterns. The red arrows mark the appearance of new diffraction peaks. The red triangle marks the diffraction peak at 9.8° which rapidly increases in intensity and becomes the strongest peak above 20.3 GPa. The blue dashed lines indicate the evolution of the sample peak positions with pressure. The yellow dashed line tracks a diffraction peak of gold (Au) which is the internal pressure standard. **b** Calculated enthalpy of the ε phase (black) relative to the δ phase (orange) as a function of pressure. **c** The Pb coordination number change from low to high pressure. The green and red bonds indicate the nine- and eight-fold coordination of Pb1 (dark gray) and Pb2 (light gray) atoms, respectively. **d** A predicted structural transition path from the initial δ to the high-pressure ε phase. The blue scissors and red bonds show the bond weakening and new bond formation, respectively.

Fig. 7) and calculations (30 GPa, Fig. 3b) may be due to the nature of the very sluggish phase transition that experimentally begins at 6.7 GPa and completes by ~80 GPa. In addition, temperature differences between the XRD experiments (~298 K) and structural searches (0 K) would in part explain the different transition pressures. Our theoretical results indicate that the δ-to-ε structural evolution involves a sequence of Pb-I bond rearrangements within the PbI₆ octahedral chains and between adjacent chains under compression (Fig. 3d, and Supplementary Figs. 15 and 16), consistent with the XRD results that suggest the system is not a simple mixing of the δ and ε phases (Fig. 3a, and Supplementary Figs. 7, 9, 12, and 13). It would rather be visualized as a set of mixed states composed of imperfect, distorted structures of δ-CsPbI₃ and ε-CsPbI₃ that keep evolving with pressure. Our calculations are unable to factor in the complex nature of the mixed phases, which further explains the underestimation of the simulated pressure for the structural and electronic transitions.

Band structure and density of state calculations were performed to study the electronic structure of the high-pressure phase. Because the low-temperature electrical transport results suggest the possibility of d-electrons serving as the charge carriers, the 5d-states of the Cs and I atoms were also included in our calculations which are close to the Cs-6s and I-5p states. At 0 GPa of δ-CsPbI₃, the VBM is mainly composed of the I-5p and Pb-6s states, while the Pb-6p and I-5p states contribute to the CBM (Fig. 4a and Supplementary

Fig. 17). The A-site Cs atom makes very little contribution to the VBM or CBM, consistent with previous results[6,28]. The involvement of d-states is negligible except for a very small contribution from the I-5d state to the upper conduction band at ~3–4 eV above the CBM. An obvious transition occurs at high pressure as the material undergoes the δ-to-ε phase transition. In the Pmn2₁ ε phase, the electronic states delocalize dramatically, and ε-CsPbI₃ has a zero bandgap at 30 GPa (Fig. 4b and Supplementary Fig. 18). If not considering the phase transition, our calculations indicate that δ-CsPbI₃ would close the bandgap at 40 GPa. These results further confirm the correlated nature of the structural and electronic transitions in compressed CsPbI₃. After examining the band structures of ε-CsPbI₃, we found that the A-site Cs cation significantly affects the electronic structure. The Cs-5d and I-5d states contribute appreciably to the valence and conduction bands, which become more pronounced with pressure up to 180 GPa (dashed lines, Fig. 4b), indicating pressure-induced electron redistribution. Quantitative orbital occupation calculations also support the pressure-induced electron redistribution (Fig. 4c). With the application of pressure, the 5d state occupation of the Cs and I atoms gradually increases, and then a sudden rise is clearly observed when the insulating δ phase transforms into the metallic ε phase (Fig. 4c). Similar electron redistribution has also been observed in the elemental form of Cs[50,51] and I[51,52] at high pressure.

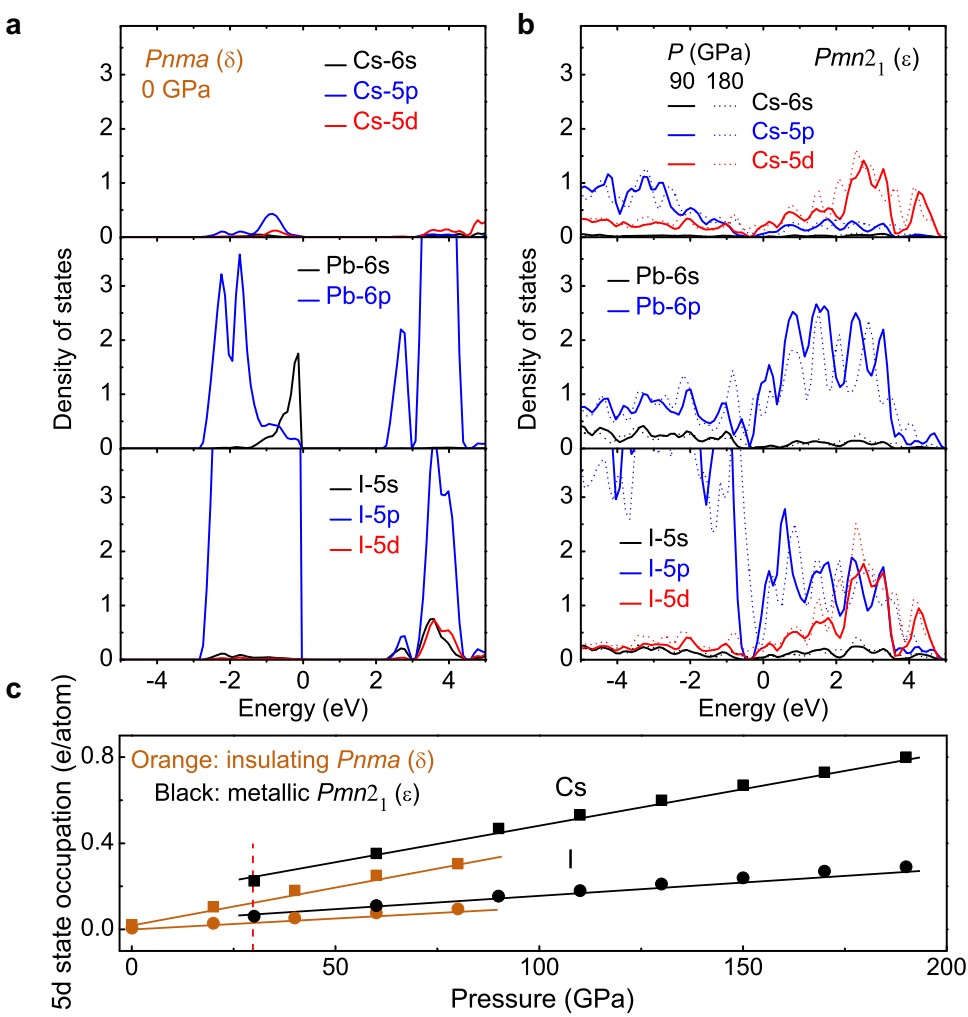

**Fig. 4 | Electronic structure of CsPbI₃ at representative pressures. a** Density of states of the δ phase at 0 GPa. The black, blue, and red lines represent the s, p, and d states, respectively. **b** Density of states of the ε phase at 90 GPa (solid lines) and 180 GPa (dashed lines). **c** The 5d state occupation of the Cs (squares) and I (circles) atoms in the initial δ (orange) and high-pressure ε (black) phases as a function of pressure. The red vertical dashed line indicates the transition pressure from the enthalpy calculations. The orange and black lines are guides for eyes.

## Discussion

The pressure-induced enhancement of the 5d state occupation explains the $\rho(T)$ - $T^n$ ($n \approx 2$) behavior in compressed CsPbI₃. Similar to material systems such as high-temperature superconductors and heavy fermion compounds where d or f electrons cause strong e-e interaction[36–39], the 5d-electrons in compressed CsPbI₃ enhance the e-e interaction and direct the electrical transport at low temperature. The observed $\rho$−$T$ curves of the metallic ε phase deviate from the predictions of a simple metal at low temperature ($\rho(T)$ ~ $T^5$), and show FL-like behavior ($\rho(T)$ ~ $AT^n$, $n \approx 2$), suggesting the presence of strong correlations. The coefficient A, which typically characterizes the strength of the e-e interaction and is related to the effective mass[53,54], also suggests strong e-e interaction (Fig. 2c). The fitted A value is $1.078 \times 10^{-8}$ Ω·cm/K$^n$ at 80 GPa, comparable with those obtained in high-temperature superconductors[54–56]. With further compression, the A value gradually decreases to $1.073 \times 10^{-9}$ Ω·cm/K$^n$ at 186 GPa.

Deviation from a FL-like state ($n < 2$) is also observed at the onset of CsPbI₃ entering the high-pressure metallic phase (Fig. 2c). Many origins for the deviation from FL behavior may be at play in correlated electron systems. Factors such as the Kondo effect[57,58] and lattice deformation and disorder[58–60] have been proposed. NFL behavior has also been observed in some unconventional superconductors and heavy fermion metals that was attributed to quantum critical

fluctuations[37,39,53]. The Kondo effect usually occurs in correlated metals doped with magnetic impurities. Magnetic impurities are not known to exist in CsPbI₃ (Supplementary Fig. 1), thus ruling out the Kondo effect as being the likely cause. It is possible that the high-pressure metallic CsPbI₃ phase has defects/disorder in the lattice. However, extrinsic disorder like grain boundaries usually increases the residual resistivity ($\rho_0$) but does not change the temperature dependence of resistivity; intrinsic disorder like lattice defects usually leads to a negative value for the coefficient A[58,59,61]. Hence, we infer that defects and disorder should not be the root cause of the deviation found in the high-pressure metallic phase. The electrical transport anomaly in compressed CsPbI₃ seems similar in character to some unconventional superconductors and heavy fermion metals, in which NFL behavior has been observed to exist for a wide range of doping levels between the under-doped and over-doped compositions that may be attributed to proximity to a nearby quantum critical point[36–39]. Pressure induces the insulator-to-metal transition and increases the carrier concentration in CsPbI₃, which resemble the evolution from an under- to an over-doped composition in unconventional superconductors and heavy fermion metals. This raises the possibility that the deviation from a FL-like state in the high-pressure metallic ε-CsPbI₃ is related to quantum criticality. In iron- and cuprate-based superconductors, such behavior usually coexists with a superconducting state in the phase diagram[36,37]. Hence,

a superconducting transition might occur below 2 K in the high-pressure metallic CsPbI$_3$. Further experiments are required to understand the origin of this anomaly and search for a superconducting phase in CsPbI$_3$ and related materials.

In summary, by compressing δ-CsPbI$_3$ to 80 GPa, the initially insulating phase transforms to a semimetallic and then to a metallic state, accompanied by the completion of a sluggish δ-to-ε phase transition from a 1D chain structure to a 3D structure that begins at 6.7 GPa and finishes at ~80 GPa. Deviation from a FL-like state is observed in the high-pressure metallic phase, which progressively evolves into a FL-like state at ~186 GPa. First-principles DFT calculations indicate that pressure induces electron redistribution which significantly enhances the 5d state occupation of Cs and I in the ε phase that strengthens the e-e interaction and eventually leads to the formation of a FL-like state. In particular, the Cs atom makes a direct and pronounced contribution to the low-temperature electrical properties of CsPbI$_3$ at high pressure. Our study opens a promising approach of manipulating the A-site cation for tuning the electronic structures and realizing intriguing electronic states in halide perovskites and analog materials at high pressures. Likewise, e-e interaction may be widely attainable in the broad family of halide perovskites, particularly at high pressure, opening routes to access a diversity of electronic states in these materials comparable to that available in the perovskite oxides.

## Methods

### Synthesis of δ-CsPbI$_3$ samples
Yellow δ-CsPbI$_3$ samples were synthesized using a solution-based method[62]. Solid PbI$_2$ (0.46 g, 1.0 mmol) and CsI (0.26 g, 1.0 mmol) powder precursors were mixed in aqueous HI (10.0 mL, 7.58 M, stabilized) in a closed 20 mL glass vial. The solution was then heated on a hotplate to 130 °C while stirring until the solid precursors were completely dissolved. The solution was then cooled to room temperature at a rate of 25–35 °C/h by turning off the hotplate while stirring the solution, yielding yellow δ-CsPbI$_3$ powders. The obtained yellow powders were collected using vacuum filtration, rinsed thoroughly with diethyl ether, and dried under reduced pressure for 12 h. The powders were then ground for half an hour before loading for experiments.

### Resistivity measurements
Resistivity measurements on yellow δ-CsPbI$_3$ at simultaneous low temperature (2–300 K) and high pressure up to 186 GPa were conducted using a nonmagnetic diamond anvil cell (made from Be-Cu alloys) with beveled diamonds with an inner and outer culet diameter of 100/300 μm in a physical property measurement system (PPMS DynaCool). PPMS DynaCool offers a high-sensitive resistivity measurement that decreases the data uncertainty, resulting in the error bars being smaller than the symbol size of data points. An insulating gasket was prepared by: (a) pre-indenting a rhenium sheet (~250 μm in thickness) to 30 μm, (b) drilling a 80-μm-diameter hole in the center of the pre-indented area using a 1064-nm laser milling machine, (c) pressing a prepared mixture of epoxy and cubic boron nitride powders (1:10 in weight) into the pre-indented area, and (d) drilling a new 40-μm-diameter hole in the center to serve as the sample chamber. Platinum foil electrodes (~4 μm in thickness) in a van der Pauw configuration were then prepared on the insulating gasket by a hand-wiring method[63]. Pressure was calibrated at room temperature using the ruby fluorescence method (0.1–30 GPa)[64] and diamond Raman peaks (30–186 GPa)[65]. No pressure transmitting medium was used to ensure good contact between the sample and the electrodes. Pressure uncertainty of 0.1 GPa and <5 GPa goes with the ruby fluorescence and diamond Raman peak methods, respectively. The pressures above 30 GPa were determined by taking the average values of the pressures at the center and the edge of the samples (Supplementary Fig. 3).

## XRD measurements
High-pressure XRD experiments were performed at beamline 12.2.2 of the Advanced Light Source, Lawrence Berkeley National Laboratory, and sector 13 of the Advanced Photon Source (APS), Argonne National Laboratory (ANL) with X-ray wavelengths of 0.4959 Å and 0.3344 Å, respectively. Powdered δ-CsPbI$_3$ samples, along with a small ruby ball (~5 μm in diameter) and a small piece of Au foil as pressure calibrants were loaded into an 80-μm-diameter (or 40-μm-diameter) sample chamber drilled in the center of a pre-indented tungsten gasket and compressed between a pair of 200-μm (or 100-μm) culet diamond anvils. Neon was loaded as the pressure transmitting medium in the 200-μm culet diamond anvil cell and no pressure medium was used in the 100-μm culet diamond anvil cell.

## DFT calculations
Structure prediction of CsPbI$_3$ at high pressure was performed via a global minimum search of the free energy surface by the swam intelligence-based CALYPSO method[66,67]. First-principles calculations included structural optimization, band structures, density of states, and orbital occupation calculations, and were carried out using DFT with the Perdew–Burke–Ernzerhof exchange-correlation functional and generalized gradient approximation[68] that are implemented in the Vienna ab initio simulation package (VASP)[69]. The projector-augmented wave pseudopotentials were set with a plane wave energy of 680 eV[70]. A fine Monkhorst-Pack Brillouin zone sampling grid with resolution of $0.04 \times 2\pi$ Å$^{-1}$ was applied. Atomic positions and lattice parameters were relaxed until all the forces on the ions were less than $10^{-3}$ eV/Å. The transition path search was performed by using the evolutionary method implemented in the CALYPSO code[71]. In our calculations, relativistic effects, e.g., spin-orbit coupling (SOC), are not included. A proof-of-principle calculation for the ε phase at 30 GPa shows that except for slight broadening of the bandwidth, density of states that includes the SOC effect is comparable with that without considering SOC and the 5d state occupation of the Cs and I atoms remains invariant (Supplementary Fig. 19).

## Data availability
All data generated or analyzed during this study are included in this article and its supplementary information files. Additional data that support the findings of this study are available from the corresponding author upon reasonable request.

## Code availability
The codes for the CALYPSO software are available at http://www.calypso.cn.

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

## Acknowledgements

This work was supported by the Department of Energy (DOE), Office of Science, Basic Energy Sciences, Materials Sciences and Engineering Division under Contract No. DE-AC02-76SF00515 (J.W., H.I.K., Y.S.L., W.L.M., and Y.L.). Beamline 12.2.2 is a DOE Office of Science User Facility under contract no. DE-AC02-05CH11231. Portions of this work were performed at GeoSoilEnviroCARS (The University of Chicago, Sector 13), APS, ANL, which is supported by the National Science Foundation – Earth Sciences (EAR – 1634415). This research used resources of the APS, a U.S. DOE Office of Science User Facility operated for the DOE Office of Science by ANL under Contract No. DE-AC02-06CH11357. N.R.W. was partially supported by a Stanford Interdisciplinary Graduate Fellowship. The simulation work was supported by the Natural Science Foundation of China under Grant No. 11904148 (K.Y.). We thank Dr. Chunjing Jia and Wen Wang for helpful discussions.

## Author contributions
F.K. and Y.L. designed the project and wrote the paper. F.K., J.Y., S.N., J.W., H.Y., Y.S.L., Y.T., W.L.M., and Y.L. conducted the experiments and analyzed the data. N.R.W synthesized the sample under H.I.K.'s supervision. K.Y. performed the calculations. All authors contributed to the discussion and revision of the paper. F.K., J.Y., and S.N. contributed equally to this work.

## Competing interests
The authors declare no competing interests.
