## [Transparent Peer Review File · Nature Communications]

Cesium-mediated electron redistribution and electron-electron interaction in high-pressure metallic CsPbI₃Reviewer #1 comments

Remarks to the Author: Overall significance

Ke et al. reported the low-temperature electrical transport study on CsPbI₃ under high pressure up to 183 GPa and discussed the Fermi liquid (FL) - like behavior. In the low temperature region, especially below the Debye-temperature, the $R(T)$ vs T^n dependence were analyzed, which yields the carrier transportation type and concentration. They reported a sluggish structure phase transition from Pnma to Pmn21, while the deviation from a FL-like state gradually disappears after entering metallic phase at 80 GPa to highest pressure 186 GPa. The A-site cation (Cs) is found to play a prominent role in the electronic structure at high pressure rather than indirect effect at ambient pressure. Overall, this work is well written and address the important issue on the low-temperature transport study in the electron-electron interaction behavior of metallic phase CsPbI₃ at high pressure. I would recommend the paper is acceptable for publication after addressing several concerns I have.

Response >> We greatly appreciate the reviewer's acknowledgement of the significance of our work and the recommendation for publication. Below we have addressed the reviewer's concerns in detail.

Remarks to the Author: Impact

The discovery of non-FL-like to FL-like behavior is the major contribution to the community in the metallic phase, which may generate a wave of research to explore similar behavior in other systems.

Response >> We thank the reviewer for highlighting the relevance and broad interest of our work.

Remarks to the Author: Strength of the claims

Concerns need to be addressed in the revision.

1) The important discovery is the n index in the $R(T)$ curve fitting at pressure 80 to 186 GPa follows a curve gradually approaching 2, the ideal FL-like state. The discussion on the origin of n deviated away from 2 is not quite clear. The XRD study shows a sluggish transition from the pnma (δ phase) to Pmn21 (ϵ phase), and completed around 82 GPa, similar pressure range as the RT resistivity behavior. The XRD data provide good evidence of δ phase pressure range, but very limit for the ϵ phase structural information (up to 93 GPa) for the non-FL-like to FL-like electronic behavior evolution. So the structural information at higher pressure XRD data would be very useful.

Response >> We have conducted an additional XRD measurement on CsPbI₃ up to 184 GPa (run 2, Fig. R1 and Supplementary Fig. 9). The experimental results indicate that no structural transition occurs between 80 and 184 GPa, that is, the non-FL-like to FL-like electronic transition progressively evolves within the high-pressure ϵ phase. The XRD patterns of the high-pressure ϵ phase in run 2 are in excellent agreement with those in run 1 (Supplementary Fig. 7). To reach up to 180 GPa, beveled diamond anvils with 100- μm culets and a thin sample were used that significantly reduce the sample diffraction volume, resulting in the overall weak intensities in the diffraction patterns in run 2. In addition, the small culet size makes the pressure control in the low-pressure regime difficult, especially with the gas membrane, so we were not able to collect as many diffraction patterns below 40 GPa. In run 2, a much shorter wavelength ($\lambda = 0.3344 \text{ \AA}$) was used versus $\lambda = 0.4959 \text{ \AA}$ in run 1, which results in the XRD patterns having relatively lower angular resolution compared to run 1. This leads to closely spaced peaks being merged in run 2, such as those peaks around the strongest reflection at $\sim 8^\circ$. Although the larger pressure step at low pressures and relatively lower resolution make it challenging to precisely determine the starting/ending pressures of the sluggish *Pnma*-to-*Pmn2₁* (δ -to- ϵ) structural transition from run 2, we are confident that the high-pressure ϵ phase persists from 80 to 184 GPa.

We have added related discussion and experimental results in the revised manuscript and supporting information.

Fig. R1. Representative XRD patterns of CsPbI₃ as a function of pressure. The three asterisks mark the background from the detector, which are pressure-invariant.

2) The claim of insulator to metal conductivity transition at 80 GPa is questionable. The R(T) between 50 GPa and 80 GPa does not show a typical insulating behavior, but rather a semi-metal. The Band structure and DOS calculation also show a significant change above 30 GPa. Please revise your claim accordingly.

Response >> We have revised the text to insulator-semimetal-metal transition as suggested.

3) This work provides a good example for the electronic property dominated by Pb-I framework at low pressure, but the cation (Cs this case) provides direct and pronounced contribution at high pressure. Since there are numerical publications on many inorganic

and organic-inorganic hybrid perovskites with similar wide band gap semiconducting to metallic phase transition, can authors extend their discovery to other systems? If so, is the non-FL-like to FL-like electronic behavior can be considered as a general mechanism or some limitation.

Response >> In our study, the Cs cation plays a direct and pronounced contribution to the electronic properties of CsPbI₃ at high pressure. Similar effects can be extended to other systems. A recent example is our own effort on expanded analogs of 3D tin-based halide perovskites with redox-active organic molecules, (dmpz)Sn₂X₆ (dmpz = N,N'-dimethylpyrazinium, X = Br⁻ and I⁻) (Angew. Chem. Int. Ed. 61, e202202911, 2022). Compression increased the conductivity of (dmpz)Sn₂X₆ to over 50 S cm⁻¹ at 60 GPa, exceeding the high-pressure conductivities of most halide perovskites. The conductivity enhancement was attributed to the increased hole density created by the A-site dmpz²⁺ reduction at high pressure. The role of the A-site cation elevates from templating the structures of 3D metal-halides to serving as charge reservoirs for tuning the carrier concentration.

Likewise, FL-like electronic behavior and electron-electron (*e-e*) interaction may be broadly attainable in halide perovskites and their derivatives, particularly at high pressure. *e-e* interaction was found to play an important role in the electronic structure and exciton density of semiconducting (FA)PbI₃ (FA = CH(NH₂)₂⁺) at ambient pressure (Phys. Rev. X 8, 021034, 2018). Pressure-induced metallization in (FA)PbI₃ was also documented in a previous report (J. Phys. Chem. Lett, 8, 2119, 2017). Such *e-e* interaction may remain in metallic (FA)PbI₃ and induce a FL-like state.

Breakdown of the FL state, however, typically requires additional factors. It is an exciting topic in the physics and materials science communities. The effect often occurs in correlated electron systems including high-temperature iron- and cuprate-based superconductors and heavy fermion compounds, and is closely related to superconductivity. However, the mechanism behind non-FL behavior is still unclear. Several factors such as the Kondo effect and quantum critical fluctuations have been proposed. It is possible to observe non-FL behavior in other halide perovskites and analogs if *e-e* interaction plays a dominant role in the low-temperature carrier transport, and magnetic impurities or quantum fluctuations exist in these materials.

In the revised manuscript, we have added more discussion to reflect the relevance of our study to a broader audience and more general systems.

4) It would be useful to see if the structure and electronic transitions are reversible upon decompression. If the high-pressure phase can be reserved to ambient condition, the FL-like behavior would be useful for ambient pressure application; if the structure is reversed back to pristine phase, it may be useful to run the compression experiment again to see any improvement in conductivity can be obtained since the contact between

nano-grains would be better.

Response >> Our XRD and electrical transport results indicate that the structural and electronic transitions are reversible upon decompression. We conducted an additional compression-decompression-recompression run (run 2 in Fig. R2 and Supplementary Fig. 5) to investigate whether the conductivity improves after pressure treatment. In run 2, upon initial compression, the resistivity of CsPbI₃ is beyond the measurable range of instrument below 20 GPa. With further compression, the resistivity falls into the measurable range, and decreases dramatically up to 90 GPa, consistent with the results in run 1. During decompression, the resistivity at ~20 GPa is comparable with that in the compression process, below which the resistivity is beyond the measurable range again, followed by the color of CsPbI₃ changing from black to red and then to yellow. During recompression, the resistivity decreases in a similar trend with the first compression cycle in run 2 and no obvious conductivity improvement is observed. We suspect that when the resistivity falls into the measurable range at 20 GPa during the first compression, the contact between nano-grains is sufficiently good and its effect on the conductivity of CsPbI₃ is minimal in the subsequent compression and decompression cycles.

Fig. R2. Room-temperature resistivity of CsPbI₃ during different experimental runs.

Remarks to the Author: Reproducibility

As mentioned in concern 4 above, it would be useful to conduct a decompression and second round compression experiment on this system to clear some uncertainty.

Response >> We conducted a decompression and second-round compression experiment for conductivity measurements (Fig. R2 & Supplementary Fig. 5). The structural and electronic transitions are reversible, and the results show excellent reproducibility.

Reviewer #2 comments

Remarks to the Author: Overall significance

In this manuscript, to illustrate the electronic structures and carrier scattering mechanisms in metal halide perovskites at low temperature combined with high pressure conditions, the authors studied electronic states in CsPbI₃ over a vast pressure-temperature space of 0.1-186 GPa and 2-300 K. The experimental results show that by compressing CsPbI₃ to 80 GPa, the insulating phase transforms to a metallic state, and a Fermi liquid-like state is observed with further compression to 186 GPa. Through the first-principles DFT calculations, the authors unravel that the Cs atom has a direct and pronounced contribution to the electrical properties of CsPbI₃ at high pressure.

Response >> We thank the reviewer's comments/suggestions to improve our manuscript. We believe we have addressed all the questions raised by the reviewer.

Remarks to the Author: Impact

This work presents an interesting strategy for tuning the electronic interaction in halide perovskites for realizing intriguing electronic states.

Response >> We greatly appreciate the reviewer's acknowledgement of the general interest and impact of our work.

Remarks to the Author: Strength of the claims

The following comments are provided to strength the claims of this work.

1) The authors mentioned that the δ -to- ϵ structural evolution involves a sequence of Pb-I bond breaking in the starting PbI₆ octahedral chains and the formation of Pb-I bonds between adjacent chains under the compression. To fully illustrate this interesting structural evolution, please provide the bond lengths and bond energies of Pb-I bonds under different structures. How does the high pressure induce the bond disconnection/connection and how about the energy transformation during this process?

Response >> We plotted the evolution of bond lengths as a function of pressure obtained from the calculations in Supplementary Fig. 16 (Fig. R3). Below we summarize the bond length evolution and the bond disconnection/connection transformation in detail. For the *Pnma* (δ) phase, there are four unique Pb-I bonds within the PbI₆ octahedron that are labeled as bond 1-4 (orange squares). Bonds 2 & 2' and bonds 3 & 3' are pairs and have the same bond length. In the plot, we also included

the distances of two pairs of non-bonded Pb and I atoms (5 and 6, red squares, dashed lines). As the structure evolves into the high-pressure ϵ phase, these two sets of atoms will form bonds together, followed by the weakening and breakage of the initial Pb-I bond 4. Therefore, five different Pb-I bond lengths are present within the PbI_8 polyhedron of $\epsilon\text{-CsPbI}_3$ (black circles, 1, 2, 3, 5 and 6). The 1 & 1', 3 & 3', and 5 & 5' bonds are pairs and have the same bond length. All the Pb-I bonds shorten with pressure and converge to a similar bond length at 93 GPa. Likewise, the PbI_9 polyhedron of $\epsilon\text{-CsPbI}_3$ also has five unique Pb-I bonds and they change in a similar fashion with those of the PbI_8 polyhedron under pressure. The evolution of the Pb-I bond length with pressure, in particular the rapid approach of the initially non-bonded Pb and I (5 and 6), aligns well with the predicted transition path. Nevertheless, we must emphasize that the calculated evolution trajectory of some atoms would be different from the actual transition path. For example, according to the calculations, the bond 4 in the initial PbI_6 octahedron progressively shortens with pressure. However, this bond will elongate and weaken during the sluggish δ -to- ϵ structural transition until it is no longer a bond in the ϵ phase. The main reason for the discrepancy is that in lattice optimization, the symmetry is prohibited from changing, while the actual structural transition is a dynamic process that constantly involves symmetry breaking and phase fraction exchange as a function of pressure.

We tried to obtain the atomic positions by fitting the XRD data and analyze the bond length change as a function of pressure. However, the δ and ϵ phases are mixed over a large pressure range (~ 77 GPa). The experimentally fitted atomic positions and bond lengths have a very large uncertainty and cannot be used to robustly constrain the variation trend as a function of pressure.

Empirically, the bond energy is inversely proportional to the bond length, i.e., the shorter the bond length, the stronger the bond. The evolution of Pb-I bond lengths with pressure suggests that the bond energies of all the Pb-I bonds increase as a function of pressure in the δ and ϵ phases. During the δ -to- ϵ structural transition, the bond energies of the 1, 2 and 3 Pb-I bonds decrease slightly and those of the 5 and 6 Pb-I bonds increase dramatically. On the contrary, the bond energy of the 4 Pb-I bond decrease significantly and they are no longer bonded in the ϵ phase. Due to the limitations of our calculations mentioned above and the complex nature of the structural transition, such evolution of bond energies is qualitative and may not fully reflect the actual energy transformation during the transition process.

Experimentally, it is challenging to measure the bond energies under compression. Instead, we tried to calculate the bond energies using DFT calculations. Bond energy (or bond enthalpy, ΔH) is typically equal to the average energy required to break apart a molecule (AB) into isolated atoms (A and B),

$$\Delta H = H(A) + H(B) - H(AB).$$

For a periodic crystal, such as $\epsilon\text{-CsPbI}_3$, it may be possible to calculate an average bond

energy of all the bonds in a unit cell by subtracting the energy of the unit cell from the energy of the isolated atoms. However, an average value of all the bonds, rather than a single bond, cannot reflect the energy transformation during the bond disconnection/connection process. Bond energy calculations at high pressure are more difficult, and no such methods have been explored. Under compression, the enthalpy ($H = U_{\text{total}} + PV$) always increases as a function of pressure because the PV term has a positive value, that is, $\Delta H = H(A) + H(B) - H(AB)$ always decreases as a function of pressure. The decrease of ΔH indicates that the average bond energy always reduces with pressure, no matter how the bond length changes, which is not consistent with the real case. These challenges hinder the quantitative study of the bond energy change and the energy transformation during the structural transition from DFT.

For clarity, we have revised bond breaking to bond weakening in the revised manuscript and supporting information and added additional discussion.

Fig. R3. Distances between Pb and neighboring I atoms as a function of pressure for the $Pnma$ (δ) and $Pmn2_1$ (ϵ) phases.

2) It is very interesting to see the sample color change with phase transition process. Is the entire phase transition process reversible? Will the color of the sample change from black to red and then to yellow by releasing the compressive force?

Response >> The structural and electronic transitions are reversible upon decompression, and the color of the sample changes from black to red and then returns to yellow after releasing pressure to ambient pressure. We have added related discussion

in the revised manuscript.

3) On page 13, the authors believed that “pressure-induced defects and disorder are not the root cause of the deviation in the high-pressure metallic phase because they are present throughout the compression process”. How do the authors know the defects and disorder exist throughout the testing process? Does the type of defect not change during the compression of the sample? If the type of defect changes, will there be an impact on the electrical transport of the sample?

Response >> Understanding the effect of defects and disorder on electrical transport properties is an interesting but challenging topic, and has been qualitatively studied in previous work (Refs 56-59 in the revised manuscript) on samples at ambient pressure. Defects and disorder may impact the electrical transport properties, but their effects on the fitted n value are usually minimal. For extrinsic disorder like grain boundaries, they increase residual resistivity (ρ_0) but do not change temperature dependence of resistivity. For intrinsic disorder like lattice defects, they usually lead to a negative value for the coefficient A (Phys. Rev. B 74, 224416, 2006 & Rep. Prog. Phys. 68, 2337, 2005). Therefore, we inferred that the effect of defects and disorder is not the root cause of the evolution of the n value from 1.75 ± 0.02 at 80 GPa to 1.98 ± 0.03 at 186 GPa in our study.

Quasi-hydrostatic pressure conditions provided by the diamond anvil cell may induce strain/stress and further defects and disorder in samples at high pressure (J. Chem. Phys. 125, 044507, 2006 & Geophys. Res. Lett. 20, 1147, 1993). It is difficult to study the evolution of defects and disorder under compression, but their effects on the electrical transport of CsPbI₃ could be similar with those at ambient pressure, that is, they may impact the electrical transport properties, but their effects on the n index are minimal. For clarity, we have revised the related content in our revised manuscript.

Remarks to the Author: Reproducibility

The reviewer noted that the starting sample for this study is the CsPbI₃ powder. The crystal orientation in the powder sample is random, which means that specific direction of the pressure applied on the CsPbI₃ cannot be determined. Dose this uncertainty influence the repeatability of testing results? Is it possible to employ the single CsPbI₃ crystal to carry out the experiments?

Response >> In our study, a diamond anvil cell was used to generate the pressure, which consists of two opposing anvils that approach each other to compress a sample enclosed in a metal gasket (Fig. R4). A hole was drilled in the center of the pre-indented gasket, severing as the sample chamber. CsPbI₃ samples, a pressure transmitting medium, and a pressure calibrant were then loaded into the sample chamber for high-

pressure measurements (see the methods for experimental details). The metal gasket provides lateral support and seals the pressure transmitting medium (in gas, liquid, or solid forms) inside the sample chamber. The pressure medium transforms the uniaxial pressure supplied by the two opposing anvils into quasi-hydrostatic pressure conditions, that is the pressures applied onto the sample are almost uniform from different directions.

Our experimental results also support that the applied pressure is quasi-hydrostatic. Our pressure calibration at different positions of samples show that the pressure gradient along the radial direction is less than 5 GPa even at 160 GPa (Supplementary Fig. 3). The samples for the XRD and transport measurements are small (less than 30 μm in diameter). Furthermore, $\delta\text{-CsPbI}_3$ is very soft and has a bulk modulus of 22.1 ± 1.2 GPa (Supplementary Fig. 13), comparable with that of NaCl (~ 24.4 GPa, one of the mostly frequently used solid pressure media that can provide quasi-hydrostatic pressure conditions). Both factors minimize the pressure gradients in the sample. We conducted multiple runs for both the XRD and conductivity measurements, and the results are repeatable (Supplementary Figs. 5, 7 and 9). More discussion about the data repeatability can be found in our responses to Reviewer 1's comments 1 and 4. Hence the raised uncertainty is minimal in our study, and does not influence the repeatability of the testing results.

We did conduct measurements on a CsPbI_3 single crystal, but the sample cracked and became polycrystalline under compression.

Fig. R4. Diamond anvil cell model.

REVIEWERS' COMMENTS

Reviewer #1 (Remarks to the Author: Overall significance):

The authors have answered most of my concerns, especially made additional XRD measurements up to 184 GPa. Here is my additional concern on the data presented in the revised version.

As the authors mentioned, the run 2 XRD were conducted with smaller culet anvils and higher energy, which gives much worse quality XRD patterns than before (Fig. S9 vs. Fig. S7) due to thinner sample and less scattering from higher energy x-ray. It looks like no much structure transition from Fig. S9, but the authors should at least process the data and fit the lattice and volume compression curves to see if there is any anomaly. The plot in right panel of Fig. S9 does not provide much information as it is impossible to distinguish small d-spacing peaks, I suggest to plot as I vs. Q.

Response >> We have analyzed the XRD data of run 2 and fitted the lattice and volume compression curves as suggested (bottom panel in Fig. R1 and Fig. R2). We find that both the lattice parameters and the unit cell volume change smoothly as a function of pressure, but they show different pressure dependence below versus above 85 GPa. The fits to a third-order Birch-Murnaghan equation of state are good if fitting the data below and above 85 GPa separately, but are extremely bad if fitting all the data together. The pressure of 85 GPa is consistent with the completion pressure of the δ -to- ϵ structural transition in run 1 (~82 GPa). The results indicate that the $Pmn2_1$ structure in the intermediate state (<85 GPa) is much more compressible than that in a pure metallic ϵ phase (>85 GPa). This implies that the intermediate state is not a simple mixing of the δ and ϵ phases, consistent with the run 1 XRD and simulation results.

From 85 to 184 GPa, the lattice parameters (bottom panel in Fig. R1) and the unit cell volume (Fig. R2) change smoothly with pressure and no obvious anomaly is observed. The large pressure step below 85 GPa in the run 2 XRD makes it challenging to reassure the abnormal change of the a lattice parameter within 62 – 82 GPa observed in run 1.

The right panel of Fig. S9 has been revised as suggested.

We have revised the related contents and figures in the revised manuscript and Supplementary Information (Supplementary Figs. 12 and 13).

Fig. R1. Lattice parameters of the *Pnma* (δ) and *Pmn2*₁ (ϵ) phases normalized to the values of the δ and ϵ phases at 0.1 GPa and 20.3 GPa, respectively.

Fig. R2. Pressure dependence of the volume per formula unit of CsPbI₃.

For the other three concerns I raised during my first review, I am satisfied with them.

Response >> We thank the reviewer for the positive comment on our responses.

Reviewer #2 (Remarks to the Author: Overall significance):

The authors have properly addressed all my concerns. Thus, the work will be acceptable for Nature Communications without change.

Response >> We appreciate the reviewer's recommendation for the publication of our work in *Nature Communications*.